Enhanced industrial text classification via hyper variational graph-guided global context integration

Zhang Geng
Hu Jianpeng mr@sues.edu.cn
School of Electronic and Electrical Engineering, Shanghai University of Engineering Science , Shanghai , Songjiang , China
Benítez-Andrades José Alberto
Electronic publication date: 2024 Jan 5
Publication date: 2024
Volume: 10
Electronic Location ID: e1788
Received 2023 Sep 29; Accepted 2023 Dec 11
Copyright: ©2024 Zhang and Hu
Copyright year: 2024
Copyright holder: Zhang and Hu
License: This is an open access article distributed under the terms of the Creative Commons Attribution License, which permits unrestricted use, distribution, reproduction and adaptation in any medium and for any purpose provided that it is properly attributed. For attribution, the original author(s), title, publication source (PeerJ Computer Science) and either DOI or URL of the article must be cited.
License URL: https://creativecommons.org/licenses/by/4.0/

Keywords: Hyper variational graph, Text information entropy matrix, Industrial applications, Capsule network

Funding: The National Key Research and Development Program of China 2020AAA0109300 This work was supported by the National Key Research and Development Program of China (2020AAA0109300). The funders had no role in study design, data collection and analysis, decision to publish, or preparation of the manuscript.

==============================
Background

Joint local context that is primarily processed by pre-trained models has emerged as a prevailing technique for text classification. Nevertheless, there are relatively few classification applications on small sample of industrial text datasets.

Methods

In this study, an approach of employing global enhanced context representation of the pre-trained model to classify industrial domain text is proposed. To achieve the application of the proposed technique, we extract primary text representations and local context information as embeddings by leveraging the BERT pre-trained model. Moreover, we create a text information entropy matrix through statistical computation, which fuses features to construct the matrix. Subsequently, we adopt BERT embedding and hyper variational graph to guide the updating of the existing text information entropy matrix. This process is subjected to iteration three times. It produces a hypergraph primary text representation that includes global context information. Additionally, we feed the primary BERT text feature representation into capsule networks for purification and expansion as well. Finally, the above two representations are fused to obtain the final text representation and apply it to text classification through feature fusion module.

Results

The effectiveness of this method is validated through experiments on multiple datasets. Specifically, on the CHIP-CTC dataset, it achieves an accuracy of 86.82% and an F1 score of 82.87%. On the CLUEEmotion2020 dataset, the proposed model obtains an accuracy of 61.22% and an F1 score of 51.56%. On the N15News dataset, the accuracy and F1 score are 72.21% and 69.06% respectively. Furthermore, when applied to an industrial patent dataset, the model produced promising results with an accuracy of 91.84% and F1 score of 79.71%. All four datasets are significantly improved by using the proposed model compared to the baselines. The evaluation result of the four dataset indicates that our proposed model effectively solves the classification problem.

Introduction

Text classification holds great promise in a multitude of disciplines, including information retrieval, digital libraries, and intelligence filtering. It is the action of automatically classifying text based on certain classification techniques or criteria, as demonstrated by  Liu et al. (2020). Although text classification is essential while building an industrial knowledge graph, there are currently few text classification models available for data in the industrial field. Early approaches to text classification were based on statistical learning techniques. The conventional method involves two steps to extract text features. First, human-made design or its inherent features are extracted from the text, such as term frequency-inversed document frequency (TF-IDF) (Aggarwal & Zhai, 2012; Ramos, 2003; Zhang, Gong & Wang, 2005). and then the classifier of the text is obtained by learning a support vector machine (SVM) (Cortes & Vapnik, 1995). Some researchers constructed the LSI (Wang & Zhang, 2006) model through SVM by leveraging the Dirichlet distribution. The Latent Dirichlet Allocation (LDA) model (Li, Sun & Zhang, 2008), which incorporates topics with Bayesian clustering algorithm to categorize texts, has produced better results in multiple classification text task. It is an appropriate approach to extract text feature information by using standard models, however these models lack useful semantic fusion feature.

The conventional models have low fault tolerance, which makes it incapable of enhancing text classification performance. Deep learning models have become one of the important methods for text feature extraction. However, most sequence deep learning models lack the extraction of local words or characters information, such as TextCNN (Shin et al., 2018; Zhang, Zhao & LeCun, 2015), TextRCNN (Wang et al., 2019) models, etc. But these models raise the issues that sequence learning cannot entirely focus on local information, especially the long-term dependencies among words in long texts. Although the LSTM and gated recurrent units (GRU) have partially fixed this issue, some features of words are less learned in long dependencies, which make it difficult to analyze long texts efficiently.

Several researchers, including Yao, Mao & Luo (2019), Zhang et al. (2020), Wu et al. (2022), Luo et al. (2023) and Zeng et al. (2023), have proposed addressing the limitations of sequence models by employing GCN models. GCN models utilize a message transmission approach to effectively handle word relationships and update graph data. Text can be approached by using two different graph types: static (Yao, Mao & Luo, 2019) and dynamic (Zhang et al., 2020) graphs. For static graph, to begin with the training task, it is necessary to establish a larger static graph that contains all the information between words or characters. Only the weights of the static graph can be modified, necessitating the creation of a text representation using static graphs that incorporate all available features. While this method produces better results for small sample datasets, it performs poorly for large sample datasets. As an alternative, another method is to construct text using a dynamic graph. Alternatively, dynamic graphs are employed to construct text representations. Unlike static graphs, dynamic graphs do not require the inclusion of all words from every text to establish static features. Instead, they dynamically generate graph features based on the text encountered during model training. While this approach is efficient and enables the modeling of lengthy texts, it suffers from lower precision and accuracy compared to the static graph approach.

Despite the benefits of current models, there are still specific issues that need to be addressed. Firstly, adding single information while creating a graph is not advisable because such frequency attributes do not accurately convey the meanings of text. Secondly, word vector-based text classification techniques only predict without additional semantic features to do the downstream tasks. Although the BERT combined with spatial feature representation solves most of the problems, it does not fuse the features expressed in the form of graphs. Thirdly, there is limited acquisition of text from single perspective, which fails to achieve the semantic unification of graph and sequence. To tackle these issues, the article proposes a dual-stream with hyper variational graph and semantic fusion model for text classification, know as HVGSFM, designed to handle medium and long texts. It combines hyper variational graph features through a fusion network to handle text classification.

In summary, this article presents the following significant contributions:

• We introduce the utilization of relative entropy statistical data information to construct the Text Information Entropy Matrix (TIEM) for graph neural networks. This approach effectively captures and expresses the graph features for sentence words, and the incorporation of a sliding window further enhances the model’s effectiveness.

• We propose the introduction of a semantic fusion unit (SFU) layer, which significantly enhances the semantic connection between two streams within the network. The trained model demonstrates remarkable improvements in this aspect.

• We enhance the interpretability of high-level text feature layers by leveraging an improved multi-layer capsule dynamic routing algorithm.

The organization of this article is as follows. Firstly, we provide a comprehensive review of related work in order to provide readers with a background understanding of the field. Next, we present in detail our proposed method, including its design principles and implementation steps. Subsequently, we describe the experiments conducted to validate the effectiveness of our method and present the experimental results and analysis. Finally, we summarize the main findings of the article and discuss future research directions.

Related Work

The sequential feature of text makes models based on sequence processing an initially considered more suitable feature extraction technique. The applicability of RNN models to sequential features led to their first application in the natural language field, particularly in text feature extraction. For text classification, some researchers proposed the TextRNN model (Liu, Qiu & Huang, 2016; Li et al., 2016a) using word vectors and LSTM encoding. Subsequently, related researchers proposed attention mechanisms (Raffel & Ellis, 2015; Chaudhari et al., 2021) to further enhance the representation of text features. Additionally, some researchers proposed encoding through bidirectional recursive attention networks, resulting in significant improvements in model accuracy. On the other hand, some scholars have also noted the problem of single word granularity and text length dependence in longer texts. In response, they proposed the HAN model (Yang et al., 2016; Xu et al., 2021), which assigns different weight values to phrases at word granularity level, combined with attention mechanisms. This approach achieved good results in classification tasks. Some researchers have also proposed an improved bat algorithm(IBANN)  (Bangyal, Ahmad & Rauf, 2019) in data classification tasks, the proposed approach modifies the standard BA by enhancing its exploitation capabilities and avoids escaping from local minima, and have a good results. There are also methods that use only Counter Propagation network (CPN) algorithm  (Bangyal, Ahmad & Abbas, 2013) to recognize offline handwritten English characters.

Recent years, the usage of convolutional neural network models has also made significant progress in natural language processing. The first proposal for CNN models in text classification was made by Zhang, Zhao & LeCun (2015). Wang et al. (2019) proposed the RCNN model which uses long short-term memory (LSTM) to traverse left and right sentence vectors, concatenates the obtained sentence vectors, performs convolutional pooling operations to form text features, and produces final classification results. Based on this model, Li & Ning (2020) added a HighWay layer (Srivastava, Greff & Schmidhuber, 2015) to improve the representation of text features. Zhao et al. (2018) hold the belief that the rich structure of text inevitably limits spatially insensitive methods and that differences between local and global information lead to different classification results. Based on this theory, a capsule neural network method for text classification has been proposed with good results. These methods use different approaches to aggregate text features. While these text processing methods yield good results, the lack of word or word spatial semantic features does not further improve the performance of the models.

Graph Neural Networks (GNNs) (Defferrard, Bresson & Vandergheynst, 2016; Kipf & Welling, 2016) have demonstrated remarkable success in text classification tasks by effectively capturing semantic relations between words. These tasks can be categorized as isomorphic or heterogeneous graph-based, depending on the structure employed. Isomorphic graph-based text classification models typically represent words in a text as a graph composed of word nodes, which is then utilized for document graph classification. TLGNN (Huang et al., 2019), TextING (Zhang et al., 2020), and HyperGAT (Ding et al., 2020) are examples of such models that establish relationships between edges and nodes using various text or word features. On the other hand, models like TextGCN (Yao, Mao & Luo, 2019), TensorGCN (Liu et al., 2020), and HeteGCN (Ragesh et al., 2021) operate on heterogeneous corpus-level graphs. These models consider text and words as nodes and employ node classification to classify unlabeled text. They employ diverse construction and processing strategies to accommodate the diverse nature of nodes and edges. However, these techniques may not be suitable for short texts with limited length, as the features tend to have less varied information, resulting in weak information correlation within the matrix.

Sequence models primarily focus on word positions and characteristics while disregarding spatial features. The attention mechanism addresses the relationship between interrelated words during training, which is why the BERT model outperforms conventional models in handing spatial relationships. XLNET combines the permutation language model and dual low attention for text feature construction. However, this computational approach is still lacking in graphical feature fusion, limiting the model’s ability to continuously improve prediction performance. Exploring the graph model provides an avenue to address graph construction by leveraging information entropy as a measure of text information quantity. The information entropy of asymmetric structure significantly enhances the semantic meaning between words. Given the numerous challenges encountered in the aforementioned model, we introduce the HVGSFM model as an alternative approach that leverages dual flow characteristics to address the issue of word feature interaction. This model encompasses the following components: the generation of word adjacency matrix features within the hypergraph, incorporating enhanced entropy properties, as well as the segmentation of the capsule network for aggregating word features. Ultimately, the semantic fusion unit (SFU) module is employed to filter the necessary features, leading to a satisfactory outcome in the field of industrial text classification.

Proposed Method

Assuming along text W=w1,w2,…,wn, where wk, k =1 , 2, …, n represents the words in the text. The task of text classification involves outputting O∈s1,s2,…,st,where si denotes the classification category. In this study, we propose a novel approach for text classification called hyper variational graph and semantic fusion model (HVGSFM), employing a dual-stream architecture. The complete HVGSFM architecture is depicted in Fig. 1 This section is structured into four parts. The first part pertains to the input layer, which incorporates both BERT and graph text information entropy matrix processing layer. The second part comprises of a dual-stream layer consisting of a hyper variational graph layer and a dynamic routing capsule layer. Next, the third part encompasses the semantic fusion layer utilized for aggregating semantic information, and it includes both graph structure information and capsule aggregate information. Finally, the fourth layer serves as the text classification output layer. In the subsequent sections, we elaborate on the specific structures of each neural network layer in the proposed HVGSFM.

Figure 1 As shown in figure, the HVGSFM architecture compasses two streams.

After the input tokens layer, the first stream began with the hyper variational graph layer, which was used for text entropy information matrix (TIEM) reconstruction and primary text feature extraction. The input to this layer was both BERT embedding and TIEM, and the output was text features along with the weights of TIEM updated by variational Gaussian distribution. The second stream involves the capsule layer, with BERT embedding as input and capsule text features as output. Next, the semantic fusion unit (SFU) was adopted to combine the features from both streams. Finally, the predictions were performed by using the last layer of the model.

Input layer

In the present section, character or word tokens with its masks, and the text information entropy matrix for inputs are presented. The subsequent statements illustrate how these elements are processed.

Word tokens processing

In this study, experiments are conducted on four distinct domain datasets. The ambiguity and uncertainty of sentences across different domains pose challenges in feature extraction. To address this, BERT Tokenizer (Devlin et al., 2019) is used with sentence pipeline segmentation to process the input level. Masks are applied to distinguish between words in the sentence. As shown in Fig. 2, the word parts are filled with ones while the padding parts are filled with zeros.

Figure 2 The graph key words input tokens.

The text information entropy matrix (TIEM)

The construction of the text information entropy matrix (TIEM) includes three steps, the details are shown as follows:

• Word frequency statistics: First, compute frequency values on the text to get the set S of words or characters, and then obtain the frequency vector F.

• Sliding window effect: To emphasize the significance of words in the weight matrix, a sliding window was adopted to cover several words in the text window, and then the frequency of the words inside the window was obtained, which is defined as Wi. The frequency was added to the word frequency in the prior step to produce the final total weight score V, which satisfies: (1) Vi=Fi+Wi,i=1,2,…,NS

Here we call the vector V as the document weight score vector.

• Text Information Entropy Matrix: The weight score between two words is constructed with improved relative entropy. The obtained text weight score vector is expressed as V. Subsequently, the adjacency matrix of the relative entropy of word features is given as: (2) Ai,j=Vj∗ log1.0+ViVj

Here we designate the adjacency matrix A as the Text Information Entropy Matrix (TIEM). Notably, the asymmetric entropy strategy utilized in generating the adjacency matrix resulted in a significant distribution difference. This refinement enabled us to effectively extract the quantity of information features among words present in the document.

Dual-stream layer

The dual-stream layer has been organically coupled with the hypergraph and capsule network. Specifically, the context words are meticulously analyzed via a hyper variational graph, which has been encoded with the BERT model in the initial phase. Subsequently, the multiple dynamic routing capsule network is applied in the second phase. These two parts together form the dual-stream layer of the model.

Hyper variational graph layer

HyperGAT (Ding et al., 2020) is a network designed for generating and processing graph structures. Due to the advantages of utilizing graph attention networks in hypergraphs for feature extraction, it was leveraged to propose hypergraph variational graph layer for text feature representation in this study. To obtain text graph features, two distinct aggregation algorithms were employed, facilitating the learning of heterogeneous high-order context representations on nodes and edges within the hypergraph. The mathematical formulation for the aforementioned process is expressed as follows: (3) hil=AGGRedgelhil−1,fjl∀ej∈ɛifjl=AGGRnodelhkl−1∀vk∈ej

ɛi denotes to the set of hyperedges which is connected to node vi , fjl is the representation of ej edge at the l layer. The function AGGRedgel aggregates all the hyperedge features of the graph to the node, and AGGRnodel function gathers the characteristics of nodes to the edge.

Node attention.

Given a specific node vi, the HyperGraph layer will learn the representation of all edges that connected to the node, whereas not all edges connected to this node have the same weight. Thus the weights are assigned through the attention mechanism, which satisfies: (4) fjl=σ∑vk∈ejαjkW1hkl−1

where σ is a nonlinear activation function, usually refers to the sigmoid function, W1 is a trainable matrix, αjk refers to the coefficient of edge ej at the node vk, which is calculated as follows: (5) αjk=expa1Tuk ∑vp∈ej expa1Tupuk=LeakyReLUW1hkl−1

Edge attention.

For all edge representations hkl−1∀vk∈ej, the attention mechanism is also used to learn the node representation of the next layer. It is formulated in the following function: (6) hil=σ∑ej∈ɛiβijW2fjl

where hil denotes the output representations of the node vi, W2 is a trainable matrix, βjk expresses the attention weight coefficients, which satisfies: (7) βij=expa2Tvj∑ep∈ɛi expa2Tvpvj=LeakyReLUW2fjl||W1hil−1.

The HyperGraph layer incorporates a two-way attention mechanism for nodes and edges, enabling it to capture high-level interactions between words while highlighting the weight distribution at different levels of granularity. In sentences, where word semantics can be ambiguous, it is crucial to appropriately maintain or eliminate different semantic information. Resolving this ambiguity requires the establishment of a relatively stable feature distribution.

To address this challenge, we propose the reconstruction of the text information entropy matrix (TIEM). Given an encoded graph, we solve the problem by sampling the feature distribution to avoid semantic ambiguity during the graph reconstruction process. Initially, we obtain an encoded feature distribution denoted as Z, serving as an intermediate variable. The relationship between the encoded feature distribution h and the matrix A(TIEM) can be expressed as follows: (8) qZH,A= ∏j=1nqzjH,AqzjH,A=Nzjμj,σj

where μ=AGGRμlH,A,σ=AGGRμlH,A represent the mean and variance of the distribution, and Z follows Gaussian distribution. Next sampling from its distribution. Since the direct sampling process does not contain gradient information, it is useless for the whole network. Thus, the result is resampled: Z = μ + ɛσ, where ɛ∼N0,1. The decoder reconstructs the graph by calculating the probability of an edge between any two nodes in the graph: (9) pAZ= ∏j=1n ∏k=1npAi,jzi,zjpA ˆi,jzi,zj=σziTzj

where A ˆ is the reconstructed TIEM, and the final feature presentation is obtained: (10) f1=conv1reluA ˆHWf2=conv2reluA ˆHW⋯fn=convnreluA ˆHWhf=f1,f2,…,fn

The variable n represents the number of filters for the convolution layer. These filters play a crucial role in enhancing fragment fusion syntactic information. Moreover, the hypergraph layer is particularly important, because it enables the mapping of text features from a word-based perspective. This ultimately results in the production of a reconstructed feature vector, which can then be subjected to further backpropagation processing. The article shows how to successfully reconstruct TIEM, so we called this layer as hyper variational graph (HVG).

Dynamic routing capsule layer

Recently, significant strides have been made in text classification with the help of capsule networks, as observed by researchers (Zhao et al., 2018; Sabour, Frosst & Hinton, 2017). Drawing inspiration from these advancements, we implement a similar approach in our own work. Specifically, we begin by extracting features using a BERT model. The resulting feature vector then undergoes routing through a primary capsule layer. Subsequently, a secondary capsule network is employed to dynamically route and weight various fragments of the text feature vector. This approach enhances the accuracy of the classification process. We construct the primary capsule layer using a multi-layer convolution, mathematically expressed as follows: (11) hv=convhB

To better visualize the proposed module, the details of secondary capsule layer are provided in Fig. 3.

Figure 3 The secondary capsule layer computation.

We introduce a summation variable, denoted by v, while W and u are treated as trainable parameters. The variable hc in the primary capsule layer is partitioned into n capsules to facilitate feature aggregation. First we initialize the value vector: (12) hv=v1,v2,…,vj,…,vn

(13) b=0,…,0

And calculation is expressed as follows: (14) ci=softmaxbiu ˆj|i=Wjivjsi= ∑iciju ˆj|ivj=gsj=sj21+sj2⋅sjsj2bij=bij+u ˆj|i⋅vj

where g(.) is a squashing nonlinear activation function used to ensure that the short vector distribution is close to 0 and the long one is close to 1. bij is an a priori relationship between ui and vj. The following routing algorithm can be used to produce the final result.

_______________________ Algorithm 1 Routing algorithm____________________________________________________________   ROUTING(  ˆ uj|i,r,l)   all capsule i in layer l and capsule j in layer (l + 1):bij ← 0.   for  r interactions do      for all capsule i in layer l:ci ← softmax(bi)      for all capsule j in layer (l + 1):sj ←∑  i  cijˆ uj|i        for all capsule j in layer (l + 1):vj ← squash(sj)      for all capsule i iin layer l and capsule j in layer (l + 1):bij ← bij + ˆ uj|i ⋅ vj     end for______________________________________________________________________________

And the number of capsule should equal to the number of the filter in HVG. Finally, the output hv feature vector is as followed: (15) hv=v1,v2,…,vj,…,vd

The capsule layer combines larger features into the final output feature vector through aggregation, and further merges them into the final classification result.

Semantic fusion layer

The semantic fusion unit is capable of integrating two separate information aspects by leveraging a structured gate, which is termed the semantic fusion unit (SFU). The gate computation is formulated by following functions: (16) fg= tanhWghg+Bgfv= tanhWvhv+Bvfu=σWuhg,hv+Bud=fu∗fv+1−fu∗fg

where hg denotes the features extracted from Hyper Variational Graph, and hv represents the ones from the capsule layer. The innovative concept of incorporating gated structures within the capsule network draws inspiration from the HighWay structure (Srivastava, Greff & Schmidhuber, 2015). This accomplished effective gating mechanism governs the incorporation of feature vector. Thus, the capsule layer is capable of yielding highly accurate and comprehensive models that outperform conventional structures.

Output layer

The output of the model is defined as the following formula: (17) pov=softmaxWo ∑i=0ndi+Bo

Finally, we obtain the distribution of predictions from the output layer. Suppose p ˆ for label, p for predicted probability, for every target labels, the output loss is described as an cross entropy loss: Lpk,p ˆk=pk⋅ logpk+1−p ˆk⋅log1−p ˆk

Thus, the loss function of our model is defined as follows: (18) L=−1N∑k=0NLpk,p ˆk

The upcoming section will give a comprehensive overview of conducting experiments.

Experiments

The results were acquired through a high-performance setup that comprised a PC with 80 GB RAM, an Intel Xeon Gold 6330 CPU, as well as a powerful 48 GB NVIDIA A40 device. The Pytorch deep learning framework was utilized for implementing the model.

Datasets

The model in this study was trained based on the following datasets that originated from different domains and languages. The datasets are listed in Table 1. The datasets came from different Internet public resources. The detailed information of the datasets are as follows:

Table 1 Dataset summary.

Dataset	Texts	Class	Avg.length	Max.length	Words	Train/test(ratio)	
CLUEEmotion2020	35,694	7	47.11	208	5,272	88.9/11.1	
CHIP-CTC	30,644	44	27.16	342	2,566	74.9/25.1	
N15News	61,218	24	18.68	155	138,229	70.0/30.0	
Notes.

The bold values indicates that compared to several ablation experiments, the HVGSFM model proposed in this article has the best F1 score and accuracy score when trained on three datasets.

• CLUEEmotion2020 (Li et al., 2016b) This dataset is emotion analysis corpus Li et al. (2016b) labeled with each sample annotated with one emotion label, and contains train, validation and test dataset. The label set is like, happiness, sadness, anger, disgust, fear and surprise for seven emotions classification.

• CHIP-CTC (Zong et al., 2021) The dataset is categorized based on the screening criteria utilized in clinical trials. It is sourced entirely from authentic clinical trial cases with the abbreviated data, and obtained from the standardized module within the public Chinese clinical website.

• N15News (Wang et al., 2022) It is generated from New York Times with 24 categories and contains both text and image information in each news. Here we use the text with body tag for news classification as test dataset.

Compared models

In our article, the HVGSFM compared with the following models, which divided into four groups:

• Baselines. This group includes two pre-trained models, BERT (Devlin et al., 2019) and XLNET (Yang et al., 2019). In the BERT model, the fine-tuning BERT model adds a perceptron layer composed of two full connection layers for classification. In the XLNET model, we still add the same perceptron as the classification layer.

• Seqs. This group of models includes four hybrid models, one is the TextCNN (Shin et al., 2018) with BERT and XLNET embedding for classification, and the other one is the CapsuleNet (Zhao et al., 2018) with BERT and XLNET embedding for classification. These two models take BERT, XLNET, etc. as the pre-trained embedding, and add a layer of TextCNN or CapsuleNet as the feature extraction layer, and added a perceptron for classification output layer.

• Graphs. This category consists of four hybrid GNN models. To compare extracted features performance, BERT and XLNET models have been fine-tuned and combined with GNNs. Additionally, a common perceptron classification layer has been added at the final layer. This group of models includes TextGCN (Yao, Mao & Luo, 2019) and HyperGAT,ding2020more as compared experiments.

• Ours. We use different pre-trained models as embedding to validate the proposed model in this group. Here we use BERT or XLNET model as embedding.

In conformity with the baselines reported in the research article, we have re-implemented the model. Utilizing BERT and XLNET pre-trained models as model embeddings, an optimal hyperparameter configuration was settled, which was subsequently adopted for all model parameters during training. Furthermore, we present evaluation task results for the dataset, which are meticulously compared against the findings obtained by the aforementioned model detailed in the article under consideration.

Experiments settings

Hyperparameter setting

In this study, grid search methodology was employed for all models to determine the optimal parameter values. For the HVGSFM, specific parameter configurations were set, and namely, a sliding window size of 4, Hyper Variational Graph layer dimension size of 150 with three layers and a dropout rate of 0.2 for each model layer. It is worth noting that the Pytorch deep learning framework was utilized to implement the HVGSFM, and as well as other models discussed in this study.

Evaluation metrics

As the AdamW (Kingma & Ba, 2014) optimization algorithm has been proven to be superior for our model, it was employed for model training purposes in this study. The algorithm was implemented at least 20 times while maintaining a learning rate of 10−5. Our classification performance was measured based on test accuracy and macro-averaged F1 score, which are commonly used metrics for evaluating the performance of text classification models.

Benchmark comparison

This section we compare baselines and analyzed the models from the following aspects: performance, parameters and ablation comparison. These three different aspects cover the advantages and disadvantages of the model in more details.

Performance comparison

In the present section, the aim is to summarize the results obtained from the models trained using multiple baselines and HVGSFM across three datasets, as listed in Tables 2 and 3. The presented results indicated that the results were significantly better, ranging from 1%–3% higher than those recorded for BERT and XLNET baseline models. Moreover, compared to various hybrid models based on pre-trained models, the HVGSFM has been approximately 1%–2% higher regarding the test set training results. Notably, the text feature construction method of hypergraph can contribute to the increase of the feature uncertainty of its text. However, the HVGSFM model still exhibits a relatively weaker advantage compared to single capsule neural networks. Furthermore, for short and medium texts datasets, such as CLUEEmotion2020 and CHIP-CTC dataset, the HVGSFM outperformed the classification model inclusive of pre-trained models by approximately 2%–3% higher than BERT and XLNET, specifically in terms of F1 scores. In brief, the HVGSFM exhibited noticeable classification performance even comparing with hybrid models, making it an exceptional model. Finally, the model’s efficiency was tested on English datasets with shorter texts such as N15News, showing promising generalization results across both Chinese and English datasets.

Table 2 The compared model results for F1 score, we compared four different group to test model.

Group	Model	CHIP-CTC	CLUEEmotion2020	N15News	
Baselines	BERT	0.8127	0.4779	0.6671	
XLNET	0.8129	0.4228	0.5132	
Seqs	TextCNN (BERT)	0.4835	0.2638	0.6281	
TextCNN (XLNET)	0.4154	0.3361	0.4956	
CapsNet (BERT)	0.8205	0.5011	0.6876	
CapsNet (XLNET)	0.8214	0.5045	0.6806	
Graphs	TextGCN (BERT)	0.4772	0.5080	0.6649	
TextGCN (XLNET)	0.5382	0.5015	0.6238	
HyperGCN (BERT)	0.8185	0.5061	0.6827	
HyperGCN (XLNET)	0.8141	0.4827	0.6539	
Ours	HVGSFM (XLNET)	0.7851	0.4954	0.6777	
HVGSFM (BERT, ours)	0.8287	0.5156	0.6906	
Notes.

The bold values indicates that compared to several ablation experiments, the HVGSFM model proposed in this article has the best F1 score and accuracy score when trained on three datasets.

Table 3 The compared model results for accuracy, we compared four different group to test model.

Group	Model	CHIP-CTC	CLUEEmotion2020	N15News	
Baselines	BERT	0.8473	0.5925	0.6933	
XLNET	0.8313	0.5633	0.5142	
Seqs	TextCNN (BERT)	0.8241	0.4317	0.6587	
TextCNN (XLNET)	0.8207	0.5668	0.5321	
CapsNet (BERT)	0.8516	0.6074	0.7165	
CapsNet (XLNET)	0.8475	0.6024	0.7065	
Graphs	TextGCN (BERT)	0.7574	0.6063	0.6912	
TextGCN (XLNET)	0.7907	0.6075	0.6629	
HyperGCN (BERT)	0.8495	0.6051	0.7011	
HyperGCN (XLNET)	0.8536	0.5937	0.6988	
Ours	HVGSFM (XLNET)	0.8561	0.5953	0.7084	
HVGSFM (BERT, ours)	0.8682	0.6122	0.7221	
Notes.

The bold values indicates that compared to several ablation experiments, the HVGSFM model proposed in this article has the best F1 score and accuracy score when trained on three datasets.

Model ablation

We conducted a comprehensive model ablation experiment to examine the effects of different models on our results. The experiment involved evaluating a single pretraining model, two individual stream models, and the HVGSF model. The findings, presented in Tables 4 and 5, demonstrate noteworthy insights.

Table 4 The model ablation results for F1 score.

Model	CHIP-CTC	CLUEEmotion2020	N15News	
BERT	0.8127	0.4779	0.6671	
BERT+HVG	0.8185	0.5061	0.6827	
BERT+CapsNet	0.8205	0.5011	0.6876	
HVGSFM (Ours)	0.8287	0.5156	0.6906	
Notes.

The bold representations show that compared to several baselines, sequence models and graph models, the HVGSFM model proposed in this article has the best F1 score and accuracy score when trained on three datasets.

Table 5 The model ablation results for accuracy.

Model	CHIP-CTC	CLUEEmotion2020	N15News	
BERT	0.8473	0.5925	0.6933	
BERT+HVG	0.8495	0.6051	0.7011	
BERT+CapsNet	0.8516	0.6074	0.7165	
HVGSFM (Ours)	0.8682	0.6122	0.7221	
Notes.

The bold representations show that compared to several baselines, sequence models and graph models, the HVGSFM model proposed in this article has the best F1 score and accuracy score when trained on three datasets.

Comparing the HVGSF model with the BERT model, we observed an improvement of approximately 1.6% in terms of performance. Moreover, in contrast to the individual channel models, the HVGSF model leverages combined text features, resulting in enhanced prediction accuracy. The analysis of the tables reveals that the single-stream models perform poorly on the test dataset. Conversely, the HVGSFM outperforms all other models across the entire dataset. However, its impact on the CHIP-CTC and N15News datasets is not as pronounced. Further analysis suggests that the presence of noise in the dataset and the inclusion of stop words tend to weaken crucial features during the construction of graph features.

Incorporating BERT text feature information into the HVGSFM compensates for the lack of inherent keyword features and leads to better performance results. Additionally, while BERT embedding captures character granularities details, it lacks specific keyword features information for the whole text. In light of this, HVGSFM addresses the issue by adding more keyword features from the context. So the HVGSFM model has been shown to be more effective than the single-stream model, it can effectively capture the semantic information of the text.

Parameter comparison

The present summary primarily delves into the influence of the number of Hyper Variational Graph (HVG) layers and TIEM slide window size on the results of model predictions. Therefore, we designed several experiments to evaluate the performance of HVGSFM.

Layer number effects.

In the previous section, we provided a detailed explanation of the multi-layer HVG structure. The number of HVG layers, denoted by N, is a crucial hyperparameter in the multi-layer reasoning structure model, as it directly affects the model’s reasoning ability. To thoroughly investigate this matter, we conducted an ablation experiment to compare and analyze the model. In particular, we created five variations of the model, corresponding to N = 1, 2, 3, 4, and 5, respectively. As shown in Fig. 4, when N was set to 1 or 2, there was a slight increase in the model’s prediction accuracy. On the other hand, setting N to 3 resulted in the highest accuracy and F1 score. However, when N was set to 4 or 5, there was a significant decrease in the results. Based on our extensive experimental data, we can conclude that N = 3 is the optimal hyperparameter for the HVG layers, thus confirming the feasibility and effectiveness of our proposed algorithm. As a result, the multi-layer hypergraph variational reasoning module HVG can accurately represent text features, thereby enhancing its capabilities in text classification.

Figure 4 (A–D) The HVG layer effects on model performance.

Window size effects.

This section aims to investigate the impact of different window sizes on matrix construction through a series of experiments. Specifically, several model experiments were conducted using window size values ranging from W = 1 to W = 6. As shown in Fig. 5, the experimental results convincingly demonstrate that the size of the window has a significant impact on the performance of the model. Moreover, for longer text classification datasets, such as the CLUEEmotions2020, the window effect is more prominent, especially when the window size is set to 4, 5, or 6. Among these, N = 4 shows the best results. Conversely, for shorter texts, the effect is less apparent. Although increasing the window size leads to some marginal improvement, the optimal situation still occurs when N = 4.

Noteworthily, the sliding window determines the range of key information covered in the sentence. Accordingly, better prediction results can be obtained by selecting an appropriate sentence length and window size, thereby verifying the effect of the window effect on the performance gain of the model.

Applications

Our research focuses on the application of HVGSFM in the classification of an industrial text dataset. To accomplish this objective, we obtained a corpus of text specifically from the industrial domain. The dataset was sourced from the Chinese National Patent Database and underwent a series of preprocessing steps to ensure its quality. Firstly, we performed deduplication to eliminate any duplicate text samples. Subsequently, we conducted a meticulous manual screening process to select text data that is relevant to the industrial domain. For the chosen text data, we employed a manual semi-supervised labeling approach, wherein corresponding labels were assigned to the text samples by human annotators. These labeled samples were then utilized for training the subsequent classification model.

Figure 5 (A–D) The effects of window size on results.

Throughout the process, we established a dataset consisting of 5,312 patent information on self-built industrial equipment, which was used for training and evaluation in the field of industrial text classification. This dataset covers a multitude of categories, and Fig. 6 depicts the distribution of each category within the dataset:

Figure 6 The distribution of the industrial dataset.

The dataset encompasses four categories, including equipment, process, material, and other. The text descriptions of industrial equipment account for a higher proportion. Based on the distribution of categories in the graph, we split it into a training dataset with 70% and a testing dataset with 30%. Furthermore, these four categories have the same proportion in both the training and testing datasets. Firstly, we train the HVGSFM using the training dataset. Then, we evaluate the performance of the model on the testing dataset and obtain the results shown as in Table 6.

Table 6 The compared model results for F1 score on Patents dataset.

Group	Model	Accuracy	F1 Score	
Baselines	BERT	0.9046	0.7149	
XLNET	0.8757	0.7035	
Seqs	TextCNN (BERT)	0.8958	0.7324	
TextCNN (XLNET)	0.8795	0.6198	
CapsNet (BERT)	0.9046	0.7465	
CapsNet (XLNET)	0.9121	0.7891	
Graphs	TextGCN (BERT)	0.9008	0.7139	
TextGCN (XLNET)	0.8721	0.6324	
HyperGCN (BERT)	0.8273	0.7715	
HyperGCN (XLNET)	0.8833	0.7398	
Ours	HVGSFM (XLNET)	0.9146	0.7711	
HVGSFM (BERT, ours)	0.9184	0.7971	
Notes.

The bold representations show that the HVGSFM model proposed in this article has the best F1 score and accuracy score compared to several baselines, sequence and graph models on the industrial text dataset.

The results indicated that there was a significant improvement compared to the BERT and XLNET baselines, with an increase of approximately 8.22% and 9.36% respectively. Compared to the hybrid model, there was an improvement of around 3% to 10%, which further suggested the model’s significant enhancement in text data classification within the industrial domain. Noteworthily, compared to internet datasets, HVGSFM exhibited a higher improvement rate in patents datasets, especially in categories such as equipment and materials, which covered a larger number of entities. This higher accuracy in classification can be attributed to the better feature extraction capability in the HVG layer. According to the results in Table 7, the impact of the BERT and XLNET pre-training models on HVGSFM is not pronounced, as evidenced by an increase of only nearly 2% in the F1 score.

Table 7 The model ablation results on patents dataset for F1 score.

Model	Accuracy	F1 Score	
BERT	0.9046	0.7149	
BERT+HVG	0.8273	0.7715	
BERT+CapsNet	0.9046	0.7465	
HVGSF (Ours)	0.9184	0.7971	
Notes.

The bold representations show that the proposed HVGSFM model has the best F1 score and accuracy score compared to several ablation experiments on the industrial text dataset.

The model ablation experiments, as presented in Table 7, reveal interesting findings. When comparing the model with only the HVG layer to the baseline, a significant improvement in accuracy was not observed. However, there was an approximate 5.66% increase in the F1 score. Since the F1 score takes into account both precision and recall, this suggests that the HVG layer plays a role in filtering multi-level constrained text features but has minimal impact on accuracy enhancement. Similarly, when comparing the model with only the capsule network layer to the baseline, the accuracy remained largely unaffected, with only a marginal increase of approximately 3.16% observed in the F1 score. However, when considering the results from both channels, there was an overall accuracy improvement of approximately 1.38% and an increase in the F1 score by approximately 8.22%. These outcomes indicate that the collaborative establishment of text classification features using the dual channel approach can lead to superior classification results to some extent.

Conclusions

This study presents the HVGSFM (hyper variational graph and semantic fusion model) as a novel approach for text feature aggregation. The HVGSFM comprises two key components: a hyper variational graph network and a capsule network. Notably, the hyper variational layer operates in conjunction with the Text Information Entropy Matrix (TIEM) to reconstruct the matrix, enabling the generation of a Gaussian distribution for word meaning level inference. Moreover, the multilayer hyper variational layers aggregate text information, such that semantic filters are formed. By processing edges and nodes within the hyper variational graph layer, the model effectively captures node feature information, thereby facilitating subsequent high-order feature processing. The semantic fusion unit (SFU) adeptly merges features from both streams, contributing to the final classification outcome. Experimental results demonstrate the superior performance and success of the model in text classification and inference, especially across diverse domains. In brief, the proposed hyper variational graph and text information entropy matrix play a crucial role in extracting text features and hold promise for other downstream NLP tasks. Future research endeavors may explore the potential of these techniques in open domain entity relationship extraction.

Supplemental Information

Supplemental Information 1 The overall code of the project, including the process of dataset preprocessing, model training, and dataset testing

The two files’ build. py, build. sh ’are used to build the dataset ’config. py’, this file is used to configure the parameters of the model Data_stat. py ”, this file is used to draw a statistical graph of the dataset Main. py ”, the main program used to train the model ’preprocess. py’, a preprocessing program used to convert the dataset into a format that the model can read ’run. sh’, dataset training script ’run_zh. sh’, dataset training script

Click here for additional data file.

Supplemental Information 2 Four datasets for model training and evaluation

CLUEEmotion2020 dataset is emotion analysis corpus labeled with each sample annotated with one emotion label, and contains train, validation and test dataset. CHIP-CTC dataset is categorized based on the screening criteria utilized in clinical trials. It is sourced entirely from authentic clinical trial cases with the abbreviated data, and obtained from the standardized module within the public Chinese clinical website. N15News dataset is generated from New York Times with 24 categories and contains both text and image information in each news. Here we use the text with body tag for news classification as test dataset. PatentsDataset corpus comes from industrial domain texts extracted from China’s national patent database, with over 3500 self-built industrial equipment patent information used for industrial field text classification.

Click here for additional data file.

Additional Information and Declarations

Competing Interests

Author Contributions

Data Availability

The authors declare there are no competing interests.

Geng Zhang conceived and designed the experiments, performed the experiments, analyzed the data, performed the computation work, prepared figures and/or tables, authored or reviewed drafts of the article, and approved the final draft.

Jianpeng Hu analyzed the data, authored or reviewed drafts of the article, and approved the final draft.

The following information was supplied regarding data availability:

The code and raw data are available in the Supplemental Files.

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
