# Peer review of "Enhanced industrial text classification via hyper variational graph-guided global context integration"

_PeerJ Computer Science, doi:10.7717/peerj-cs.1788_

## Round 0.1 · original submission · Major Revisions

Please, pay attention to the comments made by reviewer 1.
The decision is MAJOR and not MINOR revisions.

**Language Note:** The review process has identified that the English language must be improved. PeerJ can provide language editing services - please contact us at copyediting@peerj.com for pricing (be sure to provide your manuscript number and title). Alternatively, you should make your own arrangements to improve the language quality and provide details in your response letter. – PeerJ Staff

Reviewer 1 ·

Basic reporting

• Firstly, an abstract should summarize the major aspects of the entire paper:
1) A short background and current issues
2) the overall purpose of the study
3) basic methodology of your research
4) major findings as a result of your analysis
5) a brief summary of your interpretations and conclusions.
Please adjust your abstract according to this logic and make sure that the majority of your abstract is about your research, not the context. Most of your abstract states the research background, which is inappropriate. Please add more content about your research. What’s more, an abstract of a research paper is typically 200 to 300 words.
• There are many typos and grammatical errors in the manuscript. Besides, I suggest that you should have your paper’s language checked by a fluent English-speaker or an English major and carefully revised.
Please refer to the attached language report and address all the necessary issues.
Some errors are pointed below as typical, please correct all errors according to the examples.
• The paper needs a re-write in terms of presentation and write-up. Ask native English speaker to help you correct and formalize the sentences.
• As paper lacks to include the latest papers, these papers must be added-
A) Optimization of neural network using improved bat algorithm for data classification
B) Recognition of off-line isolated handwritten character using counter propagation network

• Add main contributions list as points in the Introduction section.
• Add the rest organization section at the end of the Introduction section.
• More clarifications and highlighted about the research gabs in the related works section.

Experimental design

• More clarifications and highlighted about the research gabs in the related works section.
• I feel that more explanation would be need on how the proposed method is performed.
• The Limitations of the proposed study need to be discussed before conclusion.
• Rewrite the Conclusion section to be:
• You must more clearly highlight the theoretical and practical implications of your research
• Discuss research contributions.
• -Indicate practical advantages (in at least one separate paragraph),

Validity of the findings

For the proposed Methodology, it is crucial to focus on the implementation details rather than solely presenting definitions. Justify the approach and outline the procedures

Reviewer 2 ·

Basic reporting

1.English Language Usage
The paper is written in clear and professional English overall. Some minor grammatical issues exist but do not impede understanding.

2.Literature Review
The introduction provides sufficient background on text classification and reviews relevant literature. Main techniques like TF-IDF, SVM, CNNs, RNNs, attention mechanisms, graph neural networks are covered.

3.Article Structure
The paper follows a standard structure for a machine learning research paper - introduction, related work, methods, experiments, results, conclusion. Figures and tables are appropriately used to illustrate the model architecture, results, etc.

4.Results
The results presented align with the proposed research goals and hypotheses described. Comparative evaluations demonstrate improved performance over baseline methods. The claims appear properly supported.
In summary, the paper meets basic expectations for quality and rigor based on a preliminary review.

Experimental design

1.Originality
The paper presents a new model architecture called HVGSFM for text classification. This appears to be original research and falls within the scope of a machine learning/NLP conference or journal.
2.Research Questions
The introduction highlights limitations of prior works in capturing semantic relationships for text classification, establishing the knowledge gap being addressed. The goal is clearly stated as developing a technique to integrate global context information.
3.Rigor
The experiments follow proper methodology by comparing to state-of-the-art baselines, performing ablation studies, evaluating on multiple datasets, and using standard evaluation metrics. The results support the claimed improvements. No obvious ethical issues are identified.
4.Methodology
The model architecture, algorithms, training details are provided in sufficient technical depth to reproduce the method. Some lower-level implementation specifics may need to be open-sourced if published.
In summary, the research significance is well motivated, with rigorous experiments demonstrating the efficacy of the proposed HVGSFM model. The methodology is adequately detailed. Overall the experiment design meets expectations for PeerJ.

Validity of the findings

1.Novelty and Impact
The paper’s impact is enough for PeerJ. The introduction establishes the limitations of prior works, which helps frame the value of the proposed approach. As a reviewer, I believe the method is novel based on the literature review. The impact seems meaningful but could be stated more clearly by the authors.
2.Data Availability
The authors provide the data with the submission.
3.Conclusions
The conclusion summarizes the key technical contributions like the hypergraph and capsule network components. The conclusions relate directly to the original goals stated in the introduction. The claims appear properly limited to what the results support.
In summary, the findings seem reasonably robust.

---

## Round 0.2 · accepted · Accept

The authors have addressed all of the reviewer's comments. After these changes, the manuscript has improved. Congratulations for your research and this manuscript.

Reviewer 1 ·

Basic reporting

ok

Experimental design

ok

Validity of the findings

ok

Additional comments

The paper has substantially been enhanced after the first revision. The paper needs a re-write in terms of presentation and write-up. Ask fluent English speaker to help you correct and formalize the sentences.